# Measurement of rock glacier surface change over different timescales using terrestrial laser scanning point clouds

Veit Ulrich[1], Jack G. Williams[1], Vivien Zahs[1], Katharina Anders[1,2], Stefan Hecht[3], Bernhard Höfle[1,2]

[1]3D Geospatial Data Processing Group (3DGeo), Institute of Geography, Heidelberg University, Germany

[2]Interdisciplinary Center for Scientific Computing (IWR), Heidelberg University, Germany

[3]Geomorphology and Soil Geography Research Group, Institute of Geography, Heidelberg University, Germany

*Correspondence to*: Prof. Dr. Bernhard Höfle, Institute of Geography, Heidelberg University, Im Neuenheimer Feld 368, 69120 Heidelberg, Germany (hoefle@uni-heidelberg.de).

**Abstract.** Topographic change at a given location usually results from multiple processes operating over different timescales.
However, interpretations of surface change are often based upon single values of movement, measured over a specified time period or in a single direction. This work presents a method to help separate surface change types that occur at different time scales related to the deformation of an active rock glacier, drawing on terrestrial lidar monitoring at sub-monthly intervals. To this end, we derive 3D topographic changes across the *Äußeres Hochebenkar* rock glacier in the Ötztal Alps. These changes are presented as the relative contribution of surface change during a three-week period (2018) to the annual surface change
(2017-2018). They are also separated according to the spatially variable direction perpendicular to the local rock glacier surface (using point cloud distance computation) and a single main direction of rock glacier flow, indicated by movement of individual boulders. In a 1500 m² sample area in the lower tongue section of the rock glacier, the contribution of the three-week period to the annual change perpendicular to the surface is 20 %, compared to 6 % in the direction of rock glacier flow. Viewing change in this way, our approach provides estimates of surface change in different directions that are dominant at different
times of the year. Our results demonstrate the benefit of more frequent lidar monitoring and, critically, the requirement of novel approaches to quantifying and disaggregating surface change, as a step towards rock glacier observation networks focusing on the analysis of 3D surface change over time.

## 1 Introduction

Rock glaciers play a key role in understanding the impact of changing environmental conditions on the high-mountain
cryosphere. They are bodies of unconsolidated debris which move downslope by creep of supersaturated mountain permafrost cohesive flow, creating special landforms as a visible expression (Barsch, 1992). Their deformation, i.e. change in shape and/or size, has shown sensitivity to atmospheric conditions at interannual (Roer et al., 2008; Sorg et al., 2015; Kellerer-Pirkelbauer et al., 2018) and seasonal (Delaloye et al., 2010; Kenner et al., 2017) timescales. Rising permafrost temperatures, which have been observed since the 1980s, have led to an acceleration of rock glacier movement (Kääb et al., 2007; Sorg et al., 2015).

Deformation can result from different mechanisms across the rock glacier, such as plastic deformation proximal to the accumulation zone, shearing within distinct layers, mass accumulation and thickening, frost heave, and thaw settlement (Barsch, 1996; Kääb et al., 1997; Krainer et al., 2015; Kenner et al., 2017). While certain processes tend to operate within distinct zones (Kenner et al., 2017), multiple mechanisms may occur in unison at a given point on the surface, with the resulting surface change representing superimposed expressions of these mechanisms.

Disaggregating the changes related to these deformation mechanisms represents a valuable step in interpreting how rock glaciers move, as well as what drives this movement. Approaches to support these interpretations based on *in situ* monitoring, such as the distribution of electrical resistivity tomography values (Zahs et al., 2019) or ground temperature records from boreholes (Kenner et al., 2017) are also of value. To distinguish changes at the surface, which are assumed to be in the order of a few centimeters within timescales of few weeks, 3D topographic data at high spatial resolution are required. These can be

obtained using lidar systems (e.g. Bollmann et al., 2012; Micheletti et al., 2016; Zahs et al., 2019), which provide high-accuracy and high-precision 3D measurements of a surface at centimeter-scale point spacing (generally equating to spatial resolution). Several studies have relied on 2.5D raster-based methods, such as the differencing of digital elevation models, to detect rock glacier surface change (Bollmann et al., 2012; Bollmann et al., 2015). These are limited, however, in representing changes to steep and complex morphology (Hodge et al., 2009; Sailer et al., 2014) and provide change in a single direction, typically

vertically. The Multiscale Model to Model Cloud Comparison (M3C2) algorithm (Lague et al., 2013) overcomes these limitations by computing point-wise cloud distances in a direction perpendicular to the local surface. While Zahs et al. (2019) have applied this approach to existing datasets of the rock glacier *Äußeres Hochebenkar*, the processes that cause the observed changes may induce movement along different predominant directions. Examining change in multiple directions may therefore be required in order to distinguish different types of surface change.

Repeated data acquisitions using lidar have seen increasing use in order to detect and quantify rock glacier surface change (Bollmann et al., 2012; Bollmann et al., 2015; Micheletti et al., 2016; Klug et al., 2017; Zahs et al., 2019). To date, most studies have used monitoring intervals of one year or longer. This is problematic where the aim is to increase the understanding of processes operating over shorter timescales, such as the movement of individual boulders, or the drivers of these processes, such as individual precipitation events. Ideally, monitoring intervals should be short enough to approach the timescale over

which changes occur, or the timescale of variability in external drivers; yet this remains difficult to define *in-situ* (e.g. Williams et al., 2019). The point cloud-based assessment of geomorphological activity at the *Äußeres Hochebenkar* conducted by Zahs et al. (2019) demonstrated that part of the deformation processes such as flow-induced rock glacier advance and longitudinal compression occurred throughout over a 12-year period at the rock glacier's lower tongue, although with variable magnitudes. However, it was shown that there are episodic processes as well, which may be masked by continuous deformation processes

at the timescale of a year.

We examine the benefits of interpreting 3D movement at sub-monthly intervals in relation to annual movement, here in the context of superimposed, and hence cumulative, surface changes. We quantify surface change based on movements that occur in different directions: movements normal to the surface of an active rock glacier, derived from the M3C2 algorithm, and

movements in the direction of rock glacier flow, derived from individual boulder tracking. The contribution of short-term

surface changes to annual surface changes will be derived for the first time, here as a ratio between the surface change occurring

during a three-week period and that over one year.

## 2 Study Site and Data

Our study site is the lower tongue area of the *Äußeres Hochebenkar* (Fig. 1), an active talus rock glacier located ~4.3 km SSW

of Obergurgl in the southern Ötztal Alps, Austria (46°50′ N, 11°01′ E). The rock glacier is 1550 m long and has a width of

160 m in the lower tongue area and up to 470 m in the upper part (Krainer, 2015). Situated at a NW-oriented glacial cirque,

the rock glacier is surrounded by the near-vertical slopes of Hangerer (3021 m a.s.l.) and Hochebenkamm (3149 m a.s.l.),

which are up to 300 m high. Long-term measurements have shown a continual movement of the rock glacier since

measurements started in 1938, with an increase of the surface velocity since the mid-1980s (Schneider, 1999; Kaufmann &

Ladstädter 2002; Bollmann et al., 2012; Klug et al., 2012; Hartl et al., 2016b). Ground penetrating radar (GPR) has been

applied to determine the depth of the bedrock, with estimates of mean thickness between 30-40 m (Hartl et al., 2016a).

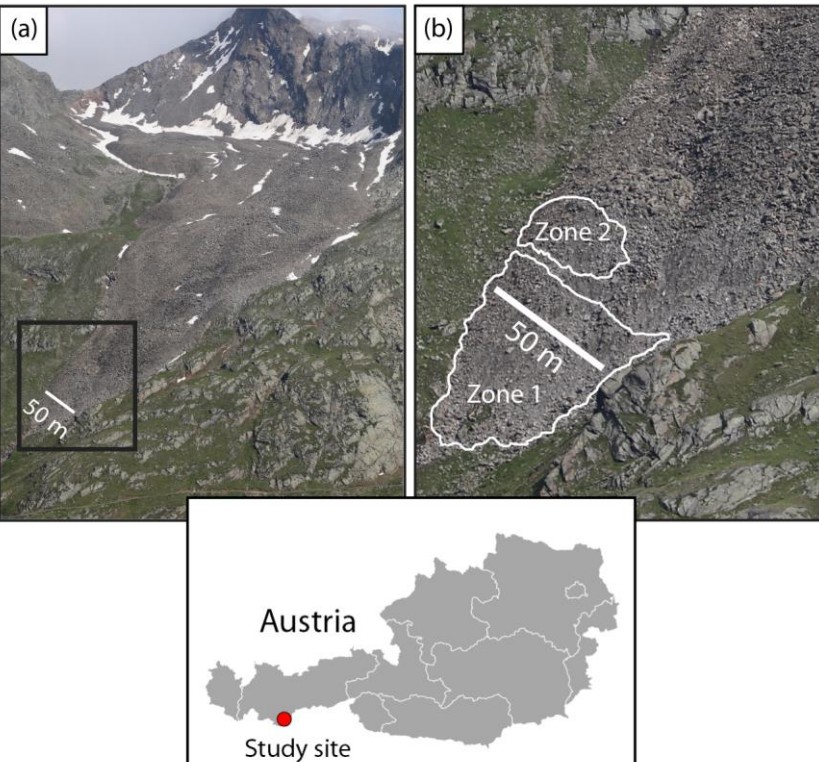

Figure 1. The *Äußeres Hochebenkar* rock glacier. (a) View of Hochebenkamm across the rock glacier (looking south) from the opposite side of the Gurgler valley. (b) Inset showing the lower tongue area. Active Zone 1 comprises the rock glacier front, while Active Zone 2 represents a ridge above the rock glacier front, which exhibits negative surface change in the

direction normal to the surface in both periods. Map data: GADM database (www.gadm.org), v.2.5, July 2015. Images: 21 July 2019.

In this paper we report on three Terrestrial Laser Scanning (TLS) datasets, captured on 19 July 2017, 7 July 2018, and 30 July 2018 from seven scan positions (six positions in the dataset of 30 July 2018), distributed around the lower tongue. Riegl VZ-

400 (acquisition in 2017) and Riegl VZ-2000i (acquisitions in 2018) TLS instruments were used, operating in the near-infrared at 1550 nm and capable of range measurement accuracies of ± 5 mm and a precision of ± 3 mm at 100 m scanning range (lower tongue width ~150 m). The measurement range over the rock glacier was up to 300 m, with accuracy and precision varying across surfaces with different target range and geometry. To obtain high point densities for an accurate representation of individual boulders, a vertical and horizontal angular resolution between 0.017° and 0.023° was chosen, which corresponds to

the maximum sampling resolution without obtaining an overlap between beams, considering the beam divergence of the TLS instrument. The resulting mean point density from all overlapping scan positions ranges from 436 points m$^{-2}$ to 528 points m$^{-2}$. Data registration accuracy was checked independently by determining the alignment error between all point clouds used in the analysis. Plane-based distances were measured between the point clouds in stable areas (rock faces in max. distance from scan positions between 11-284 m) outside the rock glacier tongue and achieved a standard deviation of residual distances of 2.4 cm

for the three-week period and 3.3 cm for the one-year period.

### 3 Methods

3D changes to the rock glacier surface were calculated using the M3C2 algorithm over 376 days (hereafter referred to as one-year) and 23 days (hereafter referred to as a three-week period), and expressed as the percentage contribution of the three-week period to the annual surface change rates (Fig. 2). The results of the M3C2 algorithm were combined with distances

between the centroids ($\mathbb{R}^3$) of corresponding boulders within each point cloud, representing movement in the direction of rock glacier flow (Fig. 3a, 2).

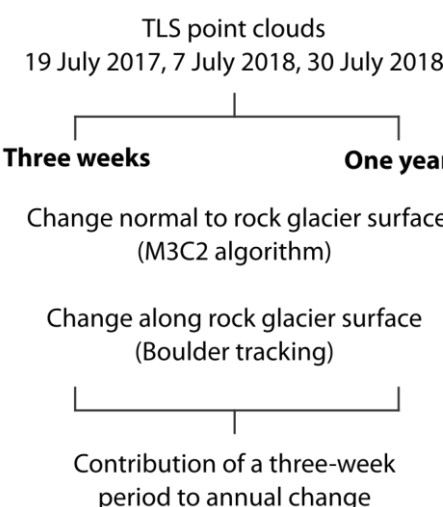

Figure 2. Workflow of the study. Changes to the rock glacier surface in the direction normal to the local surface were computed using the M3C2 algorithm over a three-week and one-year period. The results of the M3C2 algorithm were combined with distances between the centroids ($\mathbb{R}^3$) of corresponding boulders within each point cloud that represent movement in the direction of rock glacier flow. The changes in both directions are expressed as the percentage contribution of the three-week period to the annual surface change rates.

Each TLS point cloud was subsampled with a method that selects the point with the highest elevation value within a 3D spherical neighborhood of 0.05 m diameter. Using this subsampling method, no averaging was required and a consistent selection of 3D points was performed within each sphere. The uniform point distribution obtained aided the selection of a single set of parameters for the M3C2 algorithm, which was then used to quantify surface change for both periods. The method calculates signed distances between two point clouds along vectors orthogonal to the local surface, herein referred to as 'surface change in the normal direction' (Fig. 3a, 1; Lague et al., 2013). This change corresponds to the 3D distance between the average positions of two point clouds, calculated in the direction of the normal vector. The projection radius for the M3C2 algorithm, representing the volume within which average positions are calculated, was set to 1 m. In order to respect the varying roughness values of the study area, a multi-scale normal vector estimation with a minimum normal scale of 8 m was used. This was large enough to ensure that the calculated distances would not be influenced by the local orientation of single boulders, instead aligning approximately perpendicular to the rock glacier surface.

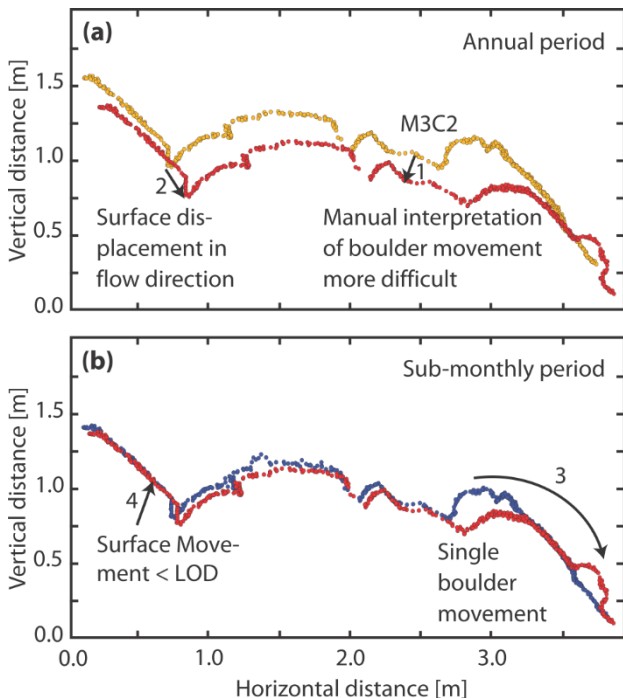

Figure 3. Schematic representation of different types of surface change that can be observed in an annual period (a) and a sub-monthly period (b). While the M3C2 distance refers to surface change in a direction normal (i.e., perpendicular) to the surface (1), the measurements of surface displacement in flow direction reflect the creep-related movement of the rock glacier tongue (2). Additionally, individual boulder movement may occur (3). In longer, annual periods, boulder movement is more difficult to identify because of overlap with creep-related surface movement. In the sub-monthly period, individual boulder movement is easier to distinguish, because creep-related changes are small and often below the level of detection (LoDetection) (4).

Various sources of uncertainty can affect the accuracy of surface changes quantified using multitemporal lidar; ultimately determining the scale of movement that can be confidently detected (Hodge et al., 2010; Lague et al., 2013; Micheletti et al., 2016). In addition to the systematic measurement errors related to the sensor and the alignment of two datasets, further uncertainty is introduced by varying point density and high roughness in morphologically complex, natural surfaces (Lague et al., 2013; Schürch et al., 2011; Soudarissanane et al, 2011). The confidence in the true position of the surface, therefore, and hence the distance to itself in successive point clouds, are spatially variable. Although the point clouds were subsampled to a uniform 0.05 m distribution, this process does not eliminate variations in surface roughness across the cloud. To account for this, the M3C2 algorithm performs a confidence assessment by approximating the minimum detectable changes, referred to as the level of detection (LoDetection). This draws on (1) the least squares fit of points within each neighborhood to a plane, with a higher standard deviation of residual distances resulting in less confidence in the surface's average position within that neighborhood, and (2) the number of points within the neighborhood, with an increase in the number of points generally

providing a more robust centroid position. The LoDetection is calculated for each point individually across the point cloud (Lague et al., 2013). In order to obtain a uniform threshold value and to make LoDetection values of both periods comparable, the 95[th] percentile of the distribution of LoDetection values was calculated for both dataset pairs used for the change analysis. For a pair of datasets, all quantified surface changes exceeding this LoDetection threshold value were considered statistically significant.

The ratio of the three-week change to the annual surface change was separated into mean positive and mean negative change. This provided an initial distinction between processes that raise or lower the surface and prevented mean values from clustering around zero where positive and negative changes were proximal. Visual inspection of the distribution of M3C2 values (Fig. 4) enabled us to identify different active zones of the rock glacier based on the direction and magnitude of surface change. In these active zones, the movement of 48 manually identified boulders with diameters ranging from 0.3 m to 1.1 m was measured

in both observed periods. For both periods, boulders rotating strongly and revealing a different geometry could not be re-identified and were not included. Active zone 1 is located at the front of the rock glacier tongue (Fig. 4), where the movement of individual boulders must not necessarily reflect rock glacier creep but may also be gravitative. In this active zone the goal of the boulder movement measurements was to separate gravitative boulder movement (Fig. 3b, 3) from boulder movement reflecting rock glacier creep. Active zone 2 is located at the top of the rock glacier body (Fig. 4). Although boulder movement

at the rock glacier surface may be influenced by processes such as frost heave or thaw settlement, causing them to move perpendicular to the rock glacier surface, their motion is predominantly in the direction of rock glacier flow (Fig. 3a, 2). The aim of the boulder movement measurements in active zone 2 was to estimate the displacement of the rock glacier in both observed periods, with the selected boulders distributed evenly across the zone. In active zone 1, it was not possible to reach an even distribution of boulders since boulders often rotate during their movement in this active zone. Here, the correspondence

between both epochs could be verified visually for a limited number of boulders only (eight in the three-week period and seven in the one-year period).

## 4 Results

The LoDetection is 0.10 m in the three-week period and 0.11 m in the one-year period ($p < 0.05$). The proportion of changes exceeding the LoDetection in the three-week period is 7.1 % as compared to 52.1 % in the one-year period. We interpret this

as relating to the low magnitudes of surface change (Table 1) relative to the surface roughness. For both the three-week and annual periods, the majority of points exhibit positive surface changes in the normal direction (68.7 % and 61.7 %, respectively), indicative of mass accumulation and thickening. Interestingly, the mean positive surface changes are 0.04 m in the three-week period (one-year period: 0.27 m) and the mean negative surface changes are -0.03 m (one-year period: -0.14 m). The contribution of the three-week period to the annual positive surface change in the normal direction amounts to 14.8 %,

while this ratio is 21.4 % for negative surface change over the same point set. The higher proportion of negative change indicates that apart from the dominant process of mass accumulation and thickening affecting both periods, surface lowering

is more active over the three-week period than surface raising. The rate of surface lowering is four times what would be expected if changes observed over the annual period were uniform through the year or, critically, if variable changes were averaged across a year by the user due to an annual survey interval (5.7 %).

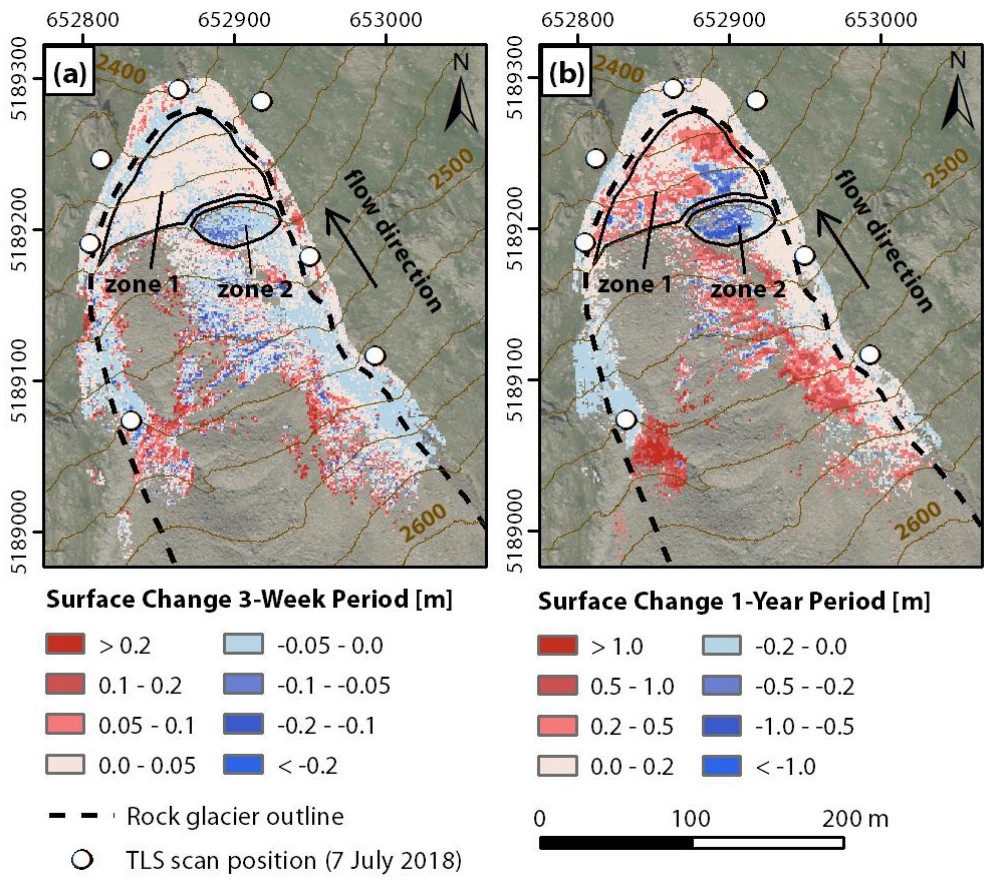

Surface Change 3-Week Period [m]

| | | | |
|---|---|---|---|
| ■ | > 0.2 | ■ | -0.05 - 0.0 |
| ■ | 0.1 - 0.2 | ■ | -0.1 - -0.05 |
| ■ | 0.05 - 0.1 | ■ | -0.2 - -0.1 |
| □ | 0.0 - 0.05 | ■ | < -0.2 |

Surface Change 1-Year Period [m]

| | | | |
|---|---|---|---|
| ■ | > 1.0 | ■ | -0.2 - 0.0 |
| ■ | 0.5 - 1.0 | ■ | -0.5 - -0.2 |
| ■ | 0.2 - 0.5 | ■ | -1.0 - -0.5 |
| □ | 0.0 - 0.2 | ■ | < -1.0 |

– – · Rock glacier outline

○ TLS scan position (7 July 2018)

Coordinate system: WGS 1984 UTM Zone 32N
Source of base map: Orthophoto Tyrol, Federal State Tyrol - data.tirol.gv.at

Figure 4. Comparison of surface changes perpendicular to local surface orientation in the lower tongue area of the *Äußeres Hochebenkar* rock glacier computed with the M3C2 algorithm for (a) the three-week period and (b) the one-year period. For a better visibility of the spatial change patterns, the scale of the color ramps is different for the two periods. Active zone 1 comprises the rock glacier front, while active zone 2 represents a ridge above the rock glacier front exhibiting negative surface change in both periods.

Surface changes over the course of a year include many small and discrete areas of positive and negative change on the orographic left side of the rock glacier front, between 2420 and 2460 m a.s.l. (active zone 1; Fig. 4b). In the three-week period,

almost no similar boulder movement is visible in a mostly static rock glacier front (Fig. 4a). Although the change detection over three weeks shows only a few significant surface changes (> 0.10 m), some similarities of surface changes can be identified in both periods (Fig. 4). For example, both periods exhibit positive surface changes in active zone 1, caused by the advance of the rock glacier, and both periods exhibit negative surface change in a ridge above the rock glacier front (active zone 2).

The ratio of distances travelled by individual boulders at the rock glacier front (active zone 1) sheds light on how independently they move relative to creep of the rock glacier. In the three-week period, only few single boulder movements (eight were detected) exceed the LoDetection, with the magnitudes of their movement differing considerably. The movement of boulders no. 1-6 in the three-week period is < 1 m (Fig. 5a), which, combined with the direction of this movement, indicates that their movement is induced by creep, reflecting the advance of the wider rock glacier body. Conversely, movement of boulders no.

7 and 8 is > 5 m (Fig. 5a), indicating that these boulders have moved gravity-induced, i.e. under their own weight, independently from the underlying material. While their movement exceeded 5 m, manual tracking of these boulders was possible within the three-week period because relatively few other observable changes occurred. This discrepancy, which is only possible from the shorter three-week monitoring interval, is important for identifying boulders moving independently from rock glacier creep.

In contrast to the three-week period, the one-year period exhibits a high quantity of single boulder movements at the rock glacier front. As this makes their re-identification between successive point clouds difficult, it is only possible to find corresponding boulders at smaller distances, whose movement is likely to result from rock glacier advance. This is indicated by magnitudes of boulder movement that are all well below 5 m (Fig. 5b). Boulders in the one-year period moving independently from the rock glacier advance (presumably over distances > 5 m) could not be identified visually, because they

likely could not be re-identified between successive point clouds due to the many occurrences and large distances of boulder movement.

Active zone 2 is an example for boulder movement induced by creep on the surface of the rock glacier (Fig. 6), which is characterized by more homogeneous movement regarding distance and direction than active zone 1. The average distance of boulder movements in the three-week period (visible as points in Fig. 6) is 0.08 m (0.023 m std. dev.) while the average

distance of boulder movements in the one-year period is 1.4 m (0.243 m std. dev.).

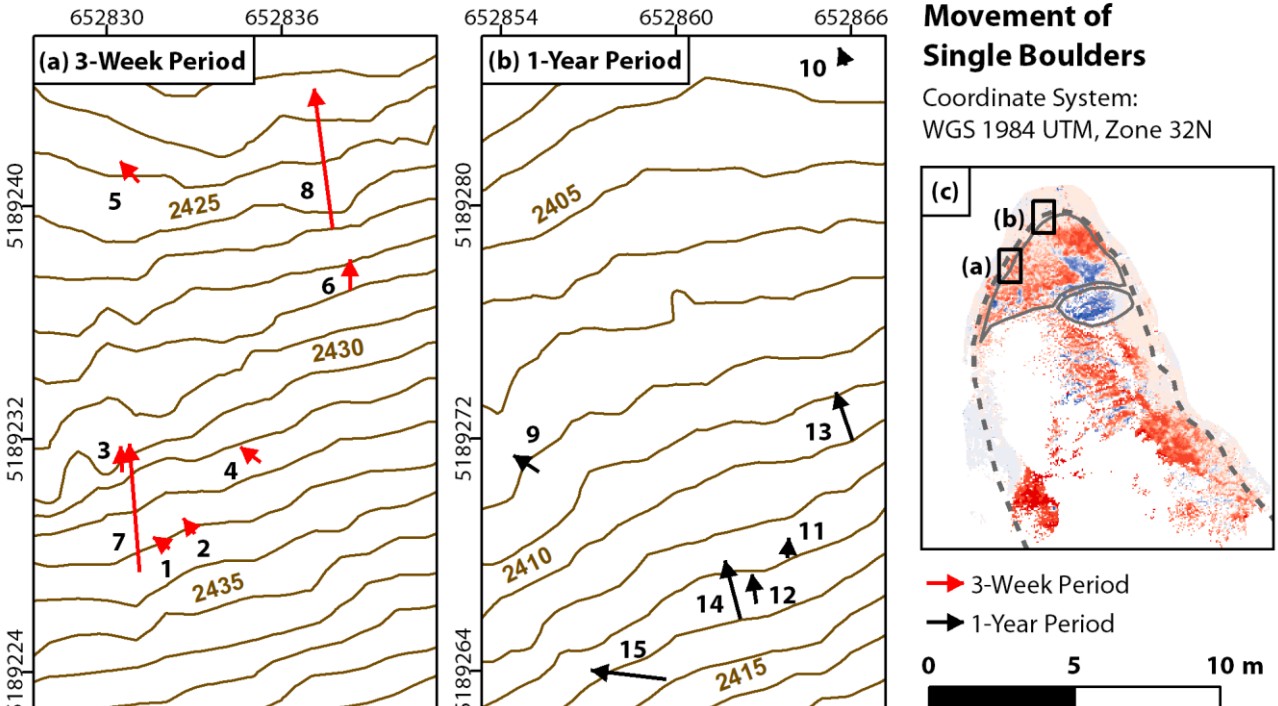

Figure 5. Movement of single boulders in two different sub-areas of the rock glacier front during the three-week period (a) and the one-year period (b). Boulders are predominantly moving in the direction of rock glacier flow. The distances covered by the moving boulders range from few cm to 6 m in the three-week period, enabling a clear distinction between creep-induced boulder movement (boulders 1 to 6) and gravity-induced boulder movement (boulders 7 and 8). In the one-year period, the magnitudes of the detected single boulder movement are more homogeneous than in the three-week period and remain well below 5 m, indicating that all of these movements are creep-induced.

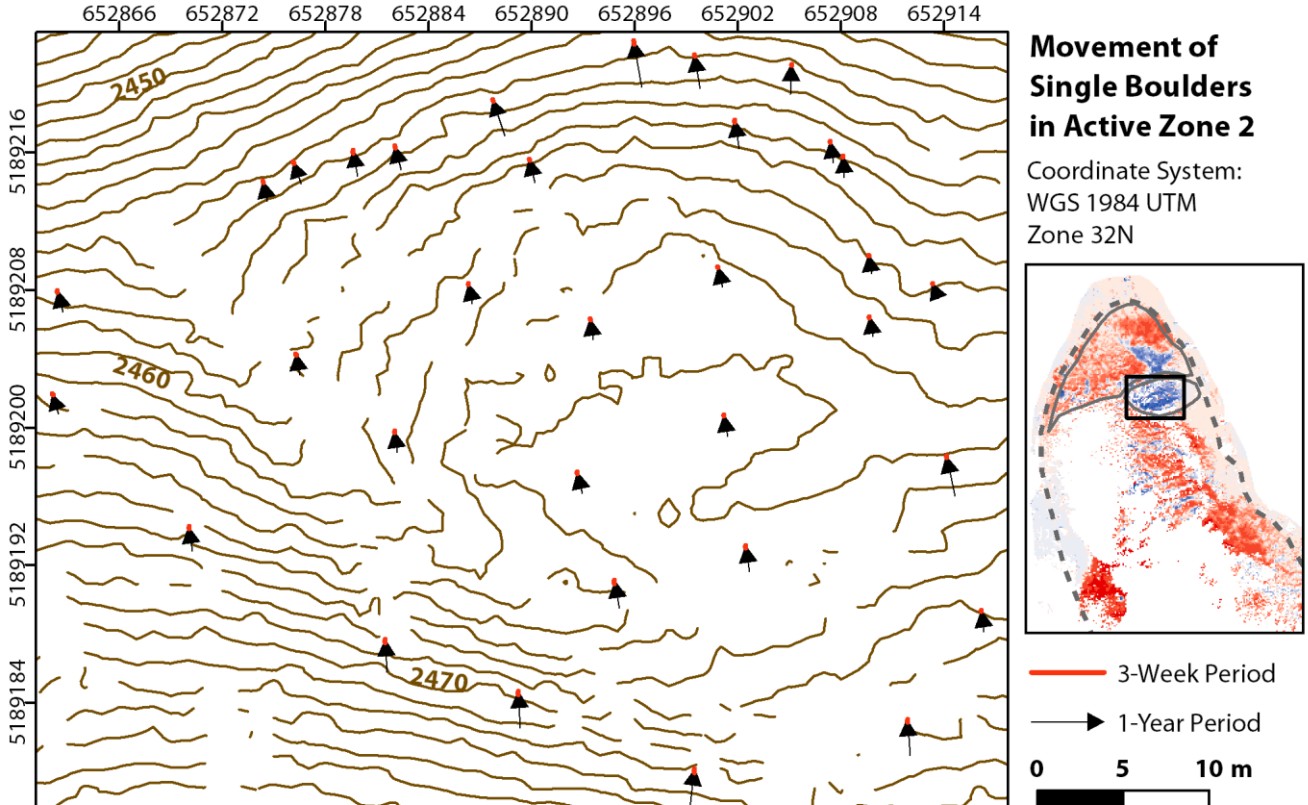

Figure 6. Movement of 33 single boulders in active zone 2 in the three-week period (red) and the one-year period (black). The boulder movements in this active zone are induced by creep, making them more homogeneous regarding distance and direction than the boulder movements in active zone 1. Because of their short distances, the boulder movements of the three-week period are only visible as points.

In comparing different change directions, the M3C2 distances illustrate that in active zone 2 (Fig. 4), the three-week period contributes -0.1 m, or 20 %, of the annual surface change (-0.5 m) in the normal direction (Fig. 7). However, the measurements of boulder movement indicate that the three-week period contributes 0.08 m, or 6 %, of the annual surface change of 1.4 m in flow direction in this active zone. Assuming a theoretical constant annual rate, the contribution of a three-week period is 5.7 %. While the movement rate in the normal direction suggests an above-average contribution of the three-week period to the annual surface change, the movement rate from the boulder movement measurements implies that quantified movement was in-line with the annual average during this three-week period.

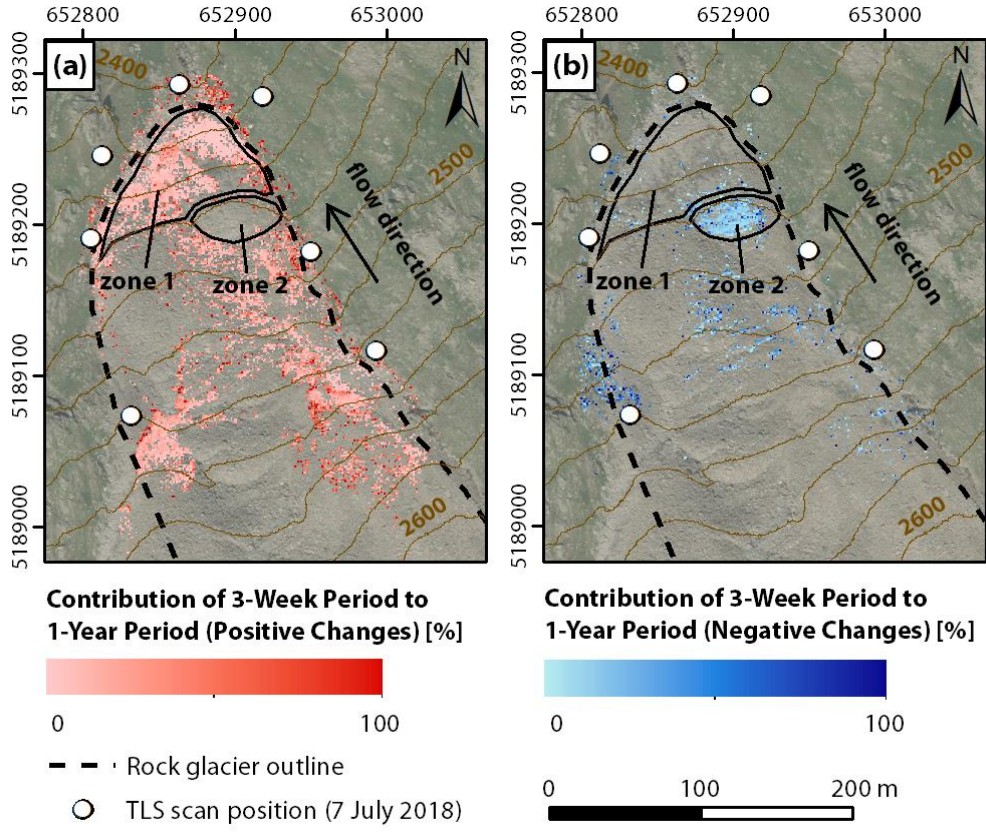

Coordinate system: WGS 1984 UTM Zone 32N
Source of base map: Orthophoto Tyrol, Federal State Tyrol - data.tirol.gv.at

Figure 7. Percentage contribution of the three-week period to the annual surface change. Active zone 1 comprises the rock glacier front, while active zone 2 represents a ridge above the rock glacier front exhibiting negative surface change in both periods. (a) positive changes, and (b) negative changes.

# 5 Discussion

## 5.1 Level of detection and implications for monitoring

Uncertainties in measuring the volume of discrete events, such as rockfalls, have been shown to accumulate with short interval monitoring (Williams et al., 2018). This occurs when single events that appear large in less frequent monitoring are in fact the sum of multiple small events, which coalesce over timescales equivalent to those of the short interval monitoring. The result is that a larger number of small events are recorded with short interval monitoring, each with a higher volumetric uncertainty relative to its size. However, this differs from displacement monitoring, where the increased proximity of the surface between scans has the effect of lowering the uncertainty in change detection, as noted by Zahs et al. (2019). Our uncertainty analysis is

consistent with these findings. We note that the identification and tracking of corresponding boulders between surveys is improved as their distance between surveys (a function of monitoring interval) lowers. However, continuous surface change mechanisms can be better recognized over the one-year period, because the portion of significant M3C2 distance values for this change detection is much higher (52.1 %) than in the three-week period (7.1 %).

Finding an appropriate temporal frequency of data capture has been identified as a key challenge to overcome the invisibility of surface change mechanisms caused by their mutual superimposition (Abellán et al., 2014). The fact that the rates of both lowering and raising of the rock glacier surface are considerably higher over three weeks as compared to the annual average shows that surface change and the drivers behind it vary seasonally. This seasonality makes it near-impossible to separate individual processes and their drivers from low-frequency monitoring. The capacity to resolve episodic processes, such as individual gravitative boulder movements at the rock glacier front, by monitoring at timescales of three weeks as compared to one year may therefore lead to the assumption that even higher monitoring frequency is desirable (e.g. daily or hourly). However, due to the small portion of significant surface changes, and the small quantity of individual gravitative boulder movement observable in a three-week period, the benefit of even more frequent monitoring may be limited for processes that are relatively slow-moving, such as rock glaciers.

Other applications involving the disaggregation of discrete changes at multiple temporal resolutions can benefit from our approach. Quantifying the contribution of a shorter time period to the changes of a longer time period and differentiating change in various directions can help to increase the understanding of the spatial and temporal distribution of coastal erosion (Westoby et al., 2018; Benjamin et al., 2020) or the development of glacial calving events (Petlicki & Kinnard 2016). Landslide analyses (e.g. Crepaldi et al., 2015) can also gain information about the local velocity of movement in different time periods by combining calculations of elevation difference with object tracking in flow direction. When applying our method, the temporal distribution must be adapted to the specific use case.

## 5.2 Implications for rock glacier understanding

Studies of other alpine rock glaciers have shown that the surface velocity reacts to seasonal temperature changes (Kääb et al., 2007; Ikeda et al., 2008; Delaloye et al., 2010). This response has been attributed to snow melt (Kääb et al., 2007) and to channeling of meltwater within the rock glacier, reducing the strength of frozen debris and promoting shearing along horizons (Ikeda et al., 2008; Kenner et al., 2017). Long-term measurements of cross-profiles on the *Äußeres Hochebenkar* rock glacier (Hartl et al., 2016) have shown that warm summers with high precipitation can lead to a decrease in surface velocity at the lower end of the tongue, indicating an ice loss due to high temperatures and a subsequent velocity decrease. Our results demonstrate that the contribution of the three-week period to the annual negative surface change in the normal direction is higher than the contribution of the three-week period to the annual positive surface change in the normal direction. This rate (as a function of annual change) is four times the rate that would be derived from a theoretical uniform rate through the course of a year (based on annual monitoring). This indicates seasonal variation in the surface change and its underlying mechanisms, including shearing or plastic deformation complemented by mechanisms involved in lowering of the surface. Given the

prevalence of surface lowering that complements down-rock glacier movement during the three-week period, it is likely that this also indicates the occurrence of thaw settlement (Kääb, 1997). The ability of our method to distinguish different seasonal

surface change processes by considering both flow direction and normal direction shows that the approach of separating different directions of surface change has considerable potential for increasing rock glacier understanding.

Rock glacier fronts are subject to material gain, caused by the advance of the rock glacier tongue (Micheletti et al., 2016). The presented surface change analysis illustrates that, in both time periods, the majority of M3C2 distances have a positive sign, indicating that the lower tongue area is affected by mass accumulation and thickening in both time periods. Multi-temporal

GPR measurements have shown that the central part of the rock glacier tongue has thinned since 2000 (Hartl et al., 2016a). At the existing cross-profiles, average surface velocities of 6.37 m a$^{-1}$ at 2570 m a.s.l. (roughly the upper end of our study area) have been measured since 1997, as compared to 1.98 m a$^{-1}$ at the lower end of the tongue (Hartl et al., 2016b). Accounting for these differences in surface velocity as well as the mass accumulation shown by our results, implies that material is being shifted from the central to the lower tongue area. This supports the theory that the lower part of the tongue is separating from

the rest of the rock glacier, as it moves into steepening terrain (~2580 m a.s.l.; Schneider & Schneider 2001).

While previous works on the surface change of rock glaciers either measured the surface velocity by tracking objects (Nickus et al., 2015; Bodin et al., 2018) or calculated changes in the normal direction with the M3C2 algorithm (Zahs et al., 2019), this study combines the M3C2 algorithm with manual measurements of boulder displacement in flow direction to separate the two directions of surface change. Seasonal variations in surface change of rock glaciers have been observed (Delaloye et al., 2010;

Kenner et al., 2017), but little is known about seasonal changes in the directions of movement. At the Muragl rock glacier in Switzerland, measurements have shown that the surface velocity increases in autumn with a time lag of approximately three months after snow melt and gradually decreases again in winter (Kääb et al., 2007), meaning that in this rock glacier, surface displacement in flow direction is dominant in autumn. This concept of different directions of surface change dominating at different times of the year illustrates that, in order to obtain a comprehensive process understanding of rock glaciers, methods

assessing change in multiple directions at each location of the rock glacier are required.

## 6 Conclusion

The aim of this study was to develop a method for multidirectional 3D change analysis drawing on terrestrial lidar monitoring at a sub-monthly interval. By considering change as the ratio of movement during a three-week period compared to the annual deformation, different surface change types related to the deformation of the lower tongue area of an active rock glacier can

be disaggregated. The analysis indicates that while the signal of continuous surface change is stronger relative to the LoDetection in a one-year period, individual boulder movements can only be resolved in the investigated three-week period. Different directions of surface change are dominant at different times of the year and can be disaggregated and estimated separately by our approach. In a sample area of the rock glacier front, the contribution of the three-week period to the annual surface change in normal direction is 20 %, while the same period only contributes 6 % to the annual surface change in the

direction of rock glacier flow as indicated by boulder movements. These findings highlight that multidirectional analyses at an increased temporal resolution (e.g. bi-weekly to monthly) will play an important role in the setup of future observation networks, because they can help to disaggregate different surface change types related to rock glacier deformation.

**Data Availability**. The data that support the findings of this study are available from the corresponding author upon reasonable
request. Multi-temporal terrestrial lidar datasets of the *Äußeres Hochebenkar* rock glacier have been openly published on PANGAEA (Pfeiffer et al., 2019).

**Author contributions**.
Conceptualization and methodology: VU, BH, JGW
Data acquisition: KA, BH, VZ
Analysis: VU
Writing the manuscript: VU, JGW, VZ, KA, SH, BH

**Competing interests**. The authors declare that they have no conflict of interest.

**Acknowledgements**. We thank two anonymous reviewers for their comments, which greatly helped to improve our manuscript. We are thankful to Professor Dietrich Barsch for his acknowledgement of the work in this study by funding the Dietrich Barsch Prize for Outstanding Students in Physical Geography.

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

**Tables**

Table 1: LoDetection threshold values and percentage of M3C2 distance values exceeding these thresholds in the one-year period and in the three-week period

|  | One-year | Three-week |
|---|---|---|
| LoDetection threshold value [m] | 0.11 | 0.10 |
| Share of M3C2 distance values exceeding the LoDetection threshold value [%] | 52.1 | 7.1 |