# Peer review of "Measurement of rock glacier surface change over different timescales using terrestrial laser scanning point clouds"

_Earth Surface Dynamics, 2020_

## Referee Comment (RC1) · Anonymous Referee #1 · 28 Jul 2020

I welcome this contribution in an interesting area of research and in the authors have previous experience. I am open to clarifying my thoughts should this be needed, the comments are well-intended, to improve the paper and not all need to be addressed but please prioritise the points of clarification raised in particular.

General comments:

The manuscript is generally well-written and the quality is potentially acceptable but there are areas of clarification needed. The multi-directional change analysis is a key area for advancing change analysis processes but could be more clearly articulated, in the abstract and better developed in the discussion.

[Figure]

Whilst the framing of the problem could be tightened up (for example, the approach presented still uses measured single values over specified time periods, which you describe as a limitation, you just add one period at higher frequency), there is value in the high temporal resolution scanning approach used, enabling disaggregation and directionality of longer term, more commonly applied annual frequency surveys. However, I cant help but feel that a simpler and much more significant addition would have been time-lapse camera imagery/photogrammetry (e.g. Blanch et al., 2020; doi:10.3390/rs12081240) over the annual period to achieve this much more comprehensively beyond a 3 week period. At the very least it would have been good to collect multiple surveys over the 3 week period to really explore the nature of the changes verses the LODs.

Links to drivers is stated as a key knowledge gap in the introduction but what were the main drivers of change over the monitored period and how discernible where they? 'individual precipitation events' (Line 54) for example or temperature variations are clearly needed to 'understand the processes operating over shorter times scales' as you state, so why have these not been used to interpret any of the changes in rates?

The fixed 3 week period (no scans in between) is extremely limited and hence prevents analysis of the surface dynamics beyond identifying that change in the different directions happens at different rates, as you would expect from a 3-dimensional deforming rock glacier surface. The final finding that ablation or downward movement is slightly higher in the warmth of July but the flow direction movement is inline with the annual average (Line 196) appears to be very intuitive. Perhaps the significance of the paper could helped by relating the spatial patterns to the drivers of change over this period as suggested in the introduction and the comment above, or by linking changes to the subsurface (you state only in the discussion at the end that there has been repeat GPR surveys – surely this would provide valuable context to the interpretation).

The interpretation of boulder movement is somewhat inconsistent. At the start of the paper they are framed as good indicators of flow direction (Line 64), then we see this

is actually more complex with a range of directions and rates (e.g. Fig 5), presumably according to their local setting (near a slope edge or interacting with other boulders or topography), but then they are still used to infer flow movement after excluding the worst 8 boulders, when presumably all may be experiencing these complexities at smaller scales?

More detail is needed on the registration errors between scans within a dataset and then between datasets. With the generally small levels of changes occurring relative to the level of detection, and the potential for the majority of the scanning scene to be creeping, which will cause issues with any ICP alignment procedures, were independent checks on data registration accuracy carried out?

Linking back to my first point, I think the paper overplays the interpretation of annual data in places, which is clearly not suitable for interpretation of seasonality, nor is it typically used that way. The discussion of this 'finding' (lines 220 – 225) is not reflective of how annual survey data are used and caution should be exercised in giving this impression. Essentially you are saying that to monitor seasonal changes you need appropriately timed surveys, not many would argue with this.

Specific comments

A clear definition of rock glaciers and their relation to debris covered glaciers for example would help.

Lines 52 – 59 need to be clarified, the implication that Zahs et al. (2019) found deformation processes occurred 'continuously' over a 12 year period is at odds with their data that showed: 'active and variable spatial and temporal surface dynamics'. Continuous movement over a 12 year period would indeed be surprising but this does not appear to be what was found, though I'm happy to be corrected by the author!

Line 60 – define 'short-interval', presumably it's application specific.

Line 61-63 the aim stated here is much clearer than in the abstract currently - consider

**ESurfD**
[Figure]

developing this for the abstract.

Fig. 1. It's unclear what the 'three epochs of TLS point clouds' are? Provide dates, as it is unclear how these relate to the changes claimed until later in the paper or switch the order and present it after line 84 and it possibly should be in the Methods section anyway. It is also unclear how you know the monitored 3 week period is one of 'heightened activity'? Is this written in retrospect or do you have more data and have selected this 3 week period as that of heightened relative activity?

Fig. 2 An inset map for context would help locate the study. Line 86: these three weeks were the only snow free conditions? Seems strange if so, and if not this does not justify why these three weeks (described in Fig. 1 as 'heightened activity').

Line 88: what were the ranges involved? Did you have to consider any range thresholds when combining the scans, what were the registration errors between scans and was the laser beam size at the maximum range relative to your sampling resolution? Was view angle/perspective (i.e. lots of very oblique measurements) an issue? These are potentially important points as you are calculating centroids of boulders – are you confident in the coverage and accuracy all around the used boulders? Are errors induced as a boulder turns, revealing a different geometry? A statement to clarifying specific error potential would help, how accurate/repeatable is the calculation of the centroids spatially across the rock glacier?

Fig. 3 Minor point, but it makes more sense to have the wider context and first survey done first – so I suggest swapping a and b around. Where is this profile on the slope? The movement of the boulder is intriguing, appearing to rise up and over a slope crest, is there an underlying ridge present? Some indication of the rock base would be very useful if available (I appreciate it may not but perhaps inferences could be made from the adjacent ridgelines). State 'Level of detection (LOD)' as you use it in the figure. (4) appears to be missing? Also I'm not sure Z – displacement is the correct title for the Y-axis as you have expressly stated M3C2 distance is not just in the Z direction, I

[Figure]

would advise this should just be 'Distance'.

Line 138 presumably the boulders were selected to represent and even coverage of the site? Comment on their distribution.

Line 140 Do boulders not interact with each other complicating this simple flow parallel movement at the centimeter scale you are considering?

Line 142 – 3 Clarify what is mean by 'The share of significant changes'

Line 148 The surface lowering is relatively more active in the 3 week period, but you also show the total amount of lowering is much lower that the amounts of accumulation – you detect mass accumulation and thickening overall, so caution should be exercised in the interpretation of these results, it might help to put these rates in absolute terms.

Line 149 what is the difference between 'rates made uniform over the year' and 'average across the year'? Furthermore, this assumption is clearly not appropriate for an alpine rock glacier which we would not expect to record uniform rates through a yearly cycle.

Fig. 4 The orientation of zones 1 and 2 are inverted and different to Fig. 2 making interpretation difficult – flow direction is now up the figure, you can work it out from the contours but it is not intuitive. Seems odd to put the map coordinate system extent int the caption but I presume this due to journal figure sizing restrictions.

Line 160 – 161 This is rather confusing. As I understand it, your 'orographic left' would be the reader's right? In this area I cant see 'small and discrete areas of positive and negative change' that you infer as 'boulder movement' but rather large (nearly half of zone 1) coherent blocks of negative (upslope) and positive (downslope) change. The change is very patchy throughout the annual change, particularly on the margins, is this registration error between scans? It seems strange to have reversed patterns of change on either side of the slope with accretion high upslope and lowering downslope in patches on the left of image 4b and the opposite in the right of zone 1, more

explanation is needed. Again perhaps combining with drivers of change or underlying bedrock might help.

Fig. 5 It would help to have similar lines in 5c denoting the glacier extent and zones to enable comparison with other figures. Not sure the hillshade background adds anything, why are the colored changes not shown to help interpretation? Why are we being shown different areas? Isn't the whole point to be able to interpret annual change better with the higher temporal resolution of the 3 week period? Therefore, we should be able to see what difference the higher frequency data make to the annual analysis.

Line 189 I'm not sure the movement data in 5a are 'more homogenous', the directions seem to differ significantly, perhaps consider rephrasing around localised but coherent directions. It seems evident that independent boulder movement must have occurred but just haven't been detected, as you point out how difficult this is in the annual data in Fig. 3b – it should be framed in this context rather than 'cannot assume they didn't'.

Line 193 What does a 0.10 m contribution to -0.5 m actually mean? Do you mean a -0.1 m contribution or did it offset the negative change by 0.1 m?

Line 234 What do you mean by an 'increase in surface velocity'? Make sure it reconciles against the final line of the results 'movement was in-line with the annual average'.

Line 248 - 255 This is the first we hear of really valuable site data. It seems a shame that this was not introduced earlier and made much more use of. Surely it is relevant to the site context and selection of zones and then interpretation of results (note the comment above on Fig. 3 for example) – and much more could be made of the spatial patterns identified if better contextualised with this previous data.

Technical corrections

Strictly speaking raster based change detection is not 2D as inferred in line 42-3, they are typically able to identify surface change in X, Y, and Z directions, and should probably be referred to as 2.5D to distinguish from full 3D analysis because the z component

is assumed normal to the raster X,Y grid (as noted in Williams et al., 2018).

Line 67: Reword 'This will enable to disclose' to better grammar.

Line 142 correct: 0.11°m

Line 171 'move creep-induced' needs rephrasing

Fig 5. Is boulder 9 the same boulder in both subsections? Presumably not so perhaps 6 is upside down?

Line 268 Should read 'at a sub-monthly interval' as you on used one three-week period.

---

## Author Comment (AC1) · 3 Aug 2020

Dear Reviewer 1,

Thank you for your constructive and thorough review, which we highly appreciate particularly during the fieldwork period and the pandemic situation. In the following, we would like to give a first response concerning your general comments. Please do not hesitate to reply and keep the discussion ongoing, and to contact us directly via email if you wish. We will address all of your specific comments and straightforward corrections in the full revision.

[Figure]

We learnt from your response and agree with your main assessment that our objective and argument with respect to drivers behind the observed change and the "causal" disaggregation of processes can be misunderstood. Our objective is to develop and explore a method for multidirectional 3D change analysis over different (overlapping) periods, which can be used for any 3D point clouds - not only LiDAR. The findings in our paper will help in developing future observation networks, where our approach centres on analysing the 3D surface change signal over time. We cannot and we do not aim in this paper to identify the underlying drivers and triggers that cause the observed, indeed potentially superimposed, surface change. We aim to separate and identify the different multidirectional change but have to leave the explanation open due to lack of long-term dense monitoring data of surface, subsurface and environment conditions. We will reformulate our paper in the full revision to clarify this element.

Bernhard Höfle and Veit Ulrich on behalf of all authors

Time-lapse imagery/photogrammetry/higher temporal resolution within three-week period

In an ideal setting we would have multiple sensors to acquire point clouds of the rock glacier that complement each other in terms of platform (i.e. diversity in viewing angle), spatial and temporal resolution, spatial coverage, direct vs. indirect georeferencing, and of course having a dense sensor network for surface, subsurface and environmental conditions (such grant proposal should be funded in the future!). The issue of how to measure surface change and how to analyse it is a crucial step that warrants deeper investigation. Surface change values are usually quickly computed with standard methods, with a meaningful understanding of the types of change they actually represent (and the drivers they could be correlated with) more difficult to gain. We therefore consider our method to be of value in creating the basis for future observation systems of rock glaciers, by covering the part of multidirectional 3D surface change analysis over different (overlapping) epochs. We would like to provide differentiated surface change data to be able to link them properly to models and other

observational data in the future. For this, it would also be interesting to achieve a higher temporal resolution and we could certainly gain additional information from time-lapse imagery/photogrammetry. With higher frequency, however, the problem of how to quantify change (over varying timescales) and how to interpret it without mixing change components remains the same. We contribute exactly to this gap. Our findings support that three weeks are a suitable monitoring interval for separating these surface changes with the proposed method (lines 225 ff.) and we do not expect a gain in information from higher frequency due to the low change rates in relation to the measurement accuracy (for LiDAR as well as photogrammetry).

Linking spatial patterns of change to drivers (precipitation etc.) and/or subsurface information

See comments above. The focus of our manuscript lies in the benefit of deriving 3D surface changes at different timescales and considering change in different directions (as opposed to only local normal direction, as is done by M3C2), which we see as a fundamental step to derive useful surface change values that can be used as input in such comprehensive and interdisciplinary rock glacier studies.

Inconsistency of boulder movement interpretation

We examine two main areas exhibiting two different types of boulder movement. At the rock glacier front (active zone 1), there is both gravitative movement of single boulders and boulder movement in flow direction, reflecting rock glacier creep. On the rock glacier surface (active zone 2), there is only boulder movement in flow direction reflecting rock glacier creep, which is why we can use boulder movements in this zone as an indicator for change in the flow direction. We agree that the distinction between these two areas, the different types of boulder movement in these areas, and the way we used the boulders as indicators for rock glacier flow is not clear and will revise these parts of the manuscript.

Overplaying of interpretation of annual data

As already outlined above, our aim is to develop a method for disaggregating surface changes (and not disaggregating and explaining the underlying drivers) using data acquired at shorter timescales. For comprehensive rock glacier monitoring, we would propose repeat measurements at the beginning and end of snow-free periods and in between (e.g. every three weeks). This would allow us to link observed changes to physical processes and their drivers.

Independent checks of data registration accuracy

Checks of data registration accuracy were carried out by measuring robust cloud-to-cloud distances between the point clouds in stable test areas (e.g. rock faces) outside the rock glacier tongue. We will include information about this in the revision.

---

## Referee Comment (RC2) · Anonymous Referee #2 · 7 Sep 2020

General comments: The overall focus of this study is to discriminate the multidirectional topographic change of an active rock glacier system in normal and flow direction and to investigate the benefits of interpreting surface change using short-term (3 weeks in July 2018) and long-term (July 2017 - July 2018) TLS time series. Interestingly, the contribution of the 3-week data to annual movement is variable. Whereas the 3-week surface change normal to the rock glacier (assessed by the M3C2 algorithm) overbalances the yearly average, creep rates of the rock glacier (assessed through boulder tracking) is in accordance with the average, long-term signal.

The topic is highly relevant for the scientific community and stimulates progress in

topographic change analysis approaches. The manuscript is written by experienced authors and largely well structured and formulated (as far as I can judge as a non-native speaker). Thanks for reading!

However, the paper needs clarification regarding several points. After careful reading and compiling a list of keywords that should be revised, improved and/or supplemented, I read the (thankfully very detailed and comprehensive!) comments of reviewer 1 as well as the authors' response. Since the vast majority of my concerns are already being questioned/discussed here, I will keep my review rather short, add some additional specific comments (see below, please ignore potential redundancies) and enter directly into the discussion.

In accordance with reviewer 1, my major concerns relate to 1) the lack of discussion on the drivers of topographic change and 2) on inconsistencies and missing method-ological details about the calculation of displacement rates in rock glacier flow direction using boulder tracking. In addition, some information is missing and some inconsisten-cies exist in the manuscript (see specific comments).

To 1.) As stated by reviewer 1, drivers of the observed (multi-temporal and multidirec-tional) topographic change are not sufficiently discussed, although they are formulated as a major objective of the study (line 61). To this request, the authors replied that the focus of the paper is strictly to develop/explore a method for multidirectional 3D change analysis over different (overlapping) periods and agreed to clarify this in the full revision of the paper. This is OK for me, although I would like to see a more detailed discussion on drivers and controls of the observed topographic change (also due to the fact that - as also already noted by Reviewer 1 - the data basis, which finally consists of two temporarily overlapping data sets, is not that extensive for a systematic methodological study). If I am right, there is many complementary (e.g. geophysical, meteorologi-cal, topoclimatic) data available for this well studied rock glacier, which might help to interpret drivers and patterns of geomorphic change in greater detail.

To 2.) As also stated by reviewer 1, the assessment of single boulder displacements remains unclear and corresponding methodological details are missing (In which areas were which boulders analysed and how?). In their reply, the authors again agreed that "the distinction between different types of boulder movement in different areas and the way how boulders are used as indicators for rock glacier flow is not clear, and that related parts of the manuscript will be revised." This is necessary and welcome.

I am looking forward to receive the revised manuscript.

Specific comments:

Chapter 1 (Introduction):

- Please provide more information on rock glaciers in general.

- Please adapt the aims of the study (drivers of topographic change).

Fig. 1:

- I suggest terming the figure "workflow of the study" and recommend shifting the figure to chapter 2.

- Change the term "epochs" and include dates.

- "Heightened activity" should be removed in this schematic diagram showing the workflow of the study (How can you know before?).

- Consider to change "boulder movement" to "boulder tracking" and "contribution to change during three-week period to annual change" to "contribution of a three-week period to annual change".

Chapter 2 (Study Site and Data):

- Please provide at least some more basic information on the (intensely studied) rock glacier (e.g., size, orientation, area, state of activity, etc.) and the study site in general.

- I recommend shifting the second paragraph with details on data acquisition (Lines

84-91) to chapter 3 (methods) and rename the chapter to "Study Site".

- Delete "therefore" (Line 90).

Fig. 2:

- Please provide an overview map.

- Integrate information on orientation in the images (e.g. "view to XY").

- Boundaries of zone 2 do not match with Fig. 4 (gap between the zones in Fig. 2b).

Chapter 3 (Methods):

- I recommend shifting information on data acquisition from chapter 2 to a first paragraph here.

- Please provide more details on data acquisition and processing (e.g. scan ranges/ footprint size, potential difficulties due to perspective/ shadowing effects, and most importantly, on registration procedure and errors (between the scan positions and between the campaigns).

Fig. 3:

- I suggest changing "Z distance" against "M3C2 distance".

- Where are the profiles located? Does the single boulder movement (3) relate to one of the boulders shown in Fig. 5a?

- Point (4) mentioned in the caption is missing in the figure - I suggest to delete this aspect in Fig. 3 and the corresponding lines 102-104 anyway (content does not refer to methodology but interpretation/ discussion).

Chapter 4 (Results):

- Line 172: I think the reference to Fig. 5b should refer to Fig. 5a, right?

- Line 175-177: How many boulders were removed from the calculation of movement

**ESurfD**
ratios (8 or 2)? Think about shifting this sentence to chapter 3 (Methods).

Fig. 5:

- Why did you choose two areas AND two periods (limited comparability)?

- Think about including information on geomorphic change instead of (or on top of) the hillshades (Fig. 5a and b).

- Please extend Fig. 5c with zone 1 and 2 from Figs. 2 and 4.

- Provide scales.

Fig. 6:

- Lines 201/202: "If changes observed over the annual period were uniform through the year or, critically, averaged across the year, the three-week period would contribute roughly 6 % to the annual change rates." I suggest to either deleting this sentence, or to add the information, that the observed 3-week contribution is higher (21.4 % for negative values and 14.8 % for positive values), if I understood correctly (Line 146-148).

- I do not understand Lines 202/203 "Red areas are relatively active during the three-week period compared to the rest of the year, while blue areas are relatively stable during the three-week period." Why are blue areas "relatively stable"? In my eyes, this contradicts the statement in line 236-238: "Our results demonstrate that the contribution of the three-week period to the annual negative surface change in the normal direction is higher than the contribution of the three-week period to the annual positive surface change in the normal direction." Please clarify.

- It would be interesting to discuss the pattern of relative change in more detail (e.g. why does the 3-week period contribute apparently much to the annual negative change in Zone 2 but not in Zone 1, where the annual data also shows strong negative values directly north of Zone 2. Probably this is due to an event that took place during the year

before the 3-week period? The positive values north of this area (see Fig. 4b) might support this idea, since the 3-week contribution to positive change in this area seems to be rather low as well.

Figs. 4, 5, 6:

- Please add north arrows to all maps (better orientation in Fig. 2).

Chapter 5 (Discussion): Part 1 - Level of detection and implications for monitoring:

- Line 212: Correct "more numerous".

- I suggest expanding the discussion with respect to different geomorphic systems, appropriate monitoring frequencies, and the benefit of differentiating multi-directional change. For an active rock glacier setting (that is typically characterized by both, "continuous" creep and "episodic" rock and boulder fall events), you presented the benefits of both, short-term TLS interval data (e.g. displacement of single boulders can be better reconstructed) and long-term data (e.g. more robust creep rates, better signal to noise ratios). This shows that a combined approach using overlapping time series and differentiated multi-directional change analyses optimizes data interpretation and the understanding of rock glacier systems. To what extent could this also apply to other geomorphic systems, or not?

Chapter 5 (Discussion): Part 2- Implications for rock glacier understanding:

- Besides the question of whether drivers of topographic change are addressed (what I would find useful), think about providing additional (mesh or raster based) volumes of mass gain and loss to get an idea about how much material is transported towards the investigated lower tongue area over time.

Chapter 5 (Discussion): Part 1 and 2:

- If the above suggestions related to the discussion are followed, I also recommend exchanging the order of the two subchapters in the discussion: Firstly implications for

rock glacier understanding and secondly (further) implications for monitoring of geo-morphic systems with respect to different monitoring intervals and the LoD.

Chapter 6 (Conclusions):

- According to the points related to the discussion, I also recommend to widen the conclusions.

---

## Author Response (AR1)

Dear Wolfgang, dear colleagues,

Thank you for taking the time to review our manuscript. Since we have already replied to the general points of reviewer 1 in our initial response and reviewer 2 has raised similar concerns, we will provide a short response to these issues followed by a more detailed reply to the specific comments. Note that given line numbers refer to the document with tracked changes.

Please do not hesitate to contact us directly via email if any questions or concerns arise,

Bernhard Höfle and Veit Ulrich on behalf of all authors

**General Points**

**Linking spatial patterns of change to drivers (precipitation etc.) and/or subsurface information**

We agree with your main assessment that our objective and argument with respect to drivers behind the observed change and the "causal" disaggregation of processes could be misunderstood. Our objective is to develop and explore a method for multidirectional 3D change analysis over different (overlapping) periods, which can be used for any 3D point clouds - not only lidar. The findings in our paper will help in developing future observation networks, where our approach centres on analysing the 3D surface change signal over time. We aim to separate and identify the changes in different directions but have to leave the explanation open due to lack of dense, long-term monitoring data of surface, subsurface and environment conditions.

To make it clearer that our focus lies on the disaggregation of different types of change rather than the assessment of underlying mechanisms, we have improved the title of the manuscript to: "Disaggregating surface change types of a rock glacier using terrestrial laser scanning point clouds acquired at different time scales"

We have adapted the abstract, introduction, and conclusion in this respect as well, in doing so also acknowledging that our work represents a step towards rock glacier observation networks focusing on the analysis of 3D surface change:

- "This work presents a method to help separate surface change types that occur at different time scales related to the deformation of an active rock glacier, drawing on terrestrial lidar monitoring at sub-monthly intervals." (lines 12 ff.)
- "Our results demonstrate the benefit of more frequent lidar monitoring and, critically, the requirement of novel approaches to quantifying and disaggregating surface change, as a step towards rock glacier observation networks focusing on the analysis of 3D surface change over time." (lines 23 ff.)
- "We examine the benefits of interpreting 3D movement at sub-monthly intervals in relation to annual movement, here in the context of superimposed, and hence cumulative, surface changes. We quantify surface change based on movements that occur in different directions: movements normal to the surface of an active rock glacier, derived from the M3C2 algorithm, and movements in the direction of rock glacier flow, derived from individual boulder tracking. The contribution of short-term surface changes to annual surface changes will be derived for the first time, here as a ratio between the surface change occurring during a three-week period and that over one year." (lines 65 ff.)

- "The aim of this study was to develop a method for multidirectional 3D change analysis drawing on terrestrial lidar monitoring at a sub-monthly interval. By considering change as the ratio of movement during a three-week period compared to the annual deformation, different surface change types related to the deformation of the lower tongue area of an active rock glacier can be disaggregated." (lines 319 ff.)
- "These findings highlight that multidirectional analyses at an increased temporal resolution (e.g. bi-weekly to monthly) will play an important role in the setup of future observation networks, because they can help to disaggregate different surface change types related to rock glacier deformation." (lines 330 ff.)

**Discussing drivers of change in more detail, adding complementary data**

The results of our paper are methods and insights that will provide suitable input for comparison and statistical analysis of rock glacier surface change components over space and time with environmental data. This will then aid the interpretation and explanation of the relation between movement and external drivers by users. The ultimate goal is that our methods can be used to establish 3D surface monitoring networks along with environmental, subsurface and wider catchment data (e.g. headwall). In our specific study case, we were working on the method development and proving that what is gained from the 3D surface analysis can be used and has increased value for further rock glacier mechanism understanding. We believe this is clearly shown in our results. We therefore improved the discussion to emphasise the "implications for rock glacier understanding" and also other geomorphological processes. Working on the drivers for the AHK rock glacier would be a different research focus and paper objective, which could make use of our disaggregated information as one important data source.

**Clearing up inconsistency of boulder movement interpretation**

We examine two main areas exhibiting two different types of boulder movement. At the rock glacier front (active zone 1), there is both gravitative movement of single boulders and boulder movement in flow direction, i.e. movement that reflects rock glacier creep. On the rock glacier surface (active zone 2), there is only boulder movement reflecting rock glacier creep, which is why we can use boulder movements in this zone as an indicator for change in the flow direction. To more clearly distinguish between these two areas, the different types of boulder movement in these areas, and also the way we use the boulders as indicators for rock glacier flow, we have added the following paragraph to the methods section:

"In these active zones, the movement of 48 manually identified boulders was measured in both observed periods. For both periods, boulders rotating strongly and revealing a different geometry could not be re-identified and were not included. Active zone 1 is located at the front of the rock glacier tongue (Fig. 4), where the movement of individual boulders must not necessarily reflect rock glacier creep but may also be gravitative. In this active zone the goal of the boulder movement measurements was to separate gravitative boulder movement (Fig. 3b, 3) from boulder movement reflecting rock glacier creep. Active zone 2 is located at the top of the rock glacier body (Fig. 4). Although boulder movement at the rock glacier surface may be influenced by processes such as frost heave or thaw settlement, causing them to move perpendicular to the rock glacier surface, their motion is predominantly in the direction of rock glacier flow (Fig. 3a, 2). The aim of the boulder movement measurements in active zone 2 was to estimate the displacement of the rock glacier in both observed periods, with the selected boulders distributed evenly across the zone. In active zone 1, it was not possible to reach an even distribution of boulders since boulders often rotate during their movement in this active zone. Here, the correspondence between both epochs could be verified visually for a limited number of boulders only." (lines 164 ff.)

**Specific comments reviewer 1**

Independent checks of data registration accuracy

We inserted information about the checks of data registration accuracy into the "study site and data" section:

"Data registration accuracy was checked independently by determining the alignment error between all point clouds used in the analysis. Plane-based distances were measured between the point clouds in stable areas (rock faces in max. distance from scan positions between 11-284 m) outside the rock glacier tongue and achieved a standard deviation of residual distances of 2.4 cm for the three-week period and 3.3 cm for the one-year period." (lines 106 ff.)

A clear definition of rock glaciers and their relation to debris covered glaciers for example would help.

We inserted a definition of rock glaciers in the beginning of the paper:

"They are bodies of unconsolidated debris which move downslope by creep of supersaturated mountain permafrost cohesive flow, creating special landforms as a visible expression (Barsch, 1992)." (lines 29 f.)

Lines 52 – 59 need to be clarified, the implication that Zahs et al. (2019) found deformation processes occurred 'continuously' over a 12 year period is at odds with their data that showed: 'active and variable spatial and temporal surface dynamics'. Continuous movement over a 12-year period would indeed be surprising but this does not appear to be what was found, though I'm happy to be corrected by the author!

To clarify that not all deformation processes are occurring continuously and to acknowledge that the magnitudes of the continual processes are variable, we changed the statement to "part of the deformation processes such as flow-induced rock glacier advance and longitudinal compression occurred throughout over a 12-year period at the rock glacier's lower tongue, although with variable magnitudes" (lines 61 f.).

Line 60 – define 'short-interval', presumably it's application specific.

We used 'short-interval' instead of 'three-week period' here, because the short interval could also be two or four weeks. We replaced the term with 'sub-monthly intervals' to show the approximate timescale we are implicating (line 65).

Line 86: these three weeks were the only snow free conditions? Seems strange if so, and if not this does not justify why these three weeks (described in Fig. 1 as 'heightened activity').

We have removed this sentence, because this three-week period was not the only snow-free period and the statement was unclear (lines 97 f.).

Line 88: what were the ranges involved? Did you have to consider any range thresholds when combining the scans, what were the registration errors between scans and was the laser beam size at the maximum range relative to your sampling resolution? Was view angle/perspective (i.e. lots of very oblique measurements) an issue? These are potentially important points as you are calculating centroids of boulders – are you confident in the coverage and accuracy all around the used boulders? Are errors induced as a boulder turns, revealing a different geometry? A statement to clarifying specific

error potential would help, how accurate/repeatable is the calculation of the centroids spatially across the rock glacier?

We included information on the ranges involved, the data acquisition settings and the registration errors of the scans:

- "The measurement range over the rock glacier was up to 300 m, with accuracy and precision varying across surfaces with different target range and geometry." (lines 101 f.)
- "To obtain high point densities for an accurate representation of individual boulders, a vertical and horizontal angular resolution between 0.017° and 0.023° was chosen, which corresponds to the maximum sampling resolution without obtaining an overlap between beams, considering the beam divergence of the TLS instrument. The resulting mean point density from all overlapping scan positions ranges from 436 points m-2 to 528 points m-2. Data registration accuracy was checked independently by determining the alignment error between all point clouds used in the analysis. Plane-based distances were measured between the point clouds in stable areas (rock faces in max. distance from scan positions between 11-284 m) outside the rock glacier tongue and achieved a standard deviation of residual distances of 2.4 cm for the three-week period and 3.3 cm for the one-year period." (lines 102 ff.)

For both periods, only boulders that could be identified in both epochs were included in the boulder tracking. These were only boulders that did not rotate much, so that their centroids hardly changed. We included a sentence concerning this in the methods section:

• "For both periods, boulders rotating strongly and revealing a different geometry could not be reidentified and were not included." (lines 165 f.)

Line 138 presumably the boulders were selected to represent and even coverage of the site? Comment on their distribution.

We added a brief description on their distribution in lines 172 ff.:

"The aim of the boulder movement measurements in active zone 2 was to estimate the displacement of the rock glacier in both observed periods, with the selected boulders distributed evenly across the zone. In active zone 1, it was not possible to reach an even distribution of boulders since boulders often rotate during their movement in this active zone. Here, the correspondence between both epochs could be verified visually for a limited number of boulders only."

Line 140 Do boulders not interact with each other complicating this simple flow parallel movement at the centimetre scale you are considering?

While interaction of boulders with each other while moving as part of creep on the rock glacier surface cannot be fully excluded, we observe relative positions of boulders to remain similar and therefore consider interaction between them to be negligible for the detected changes.

Line 142 – 3 Clarify what is meant by 'The share of significant changes'

**We adapted the phrase to "The proportion of changes exceeding the level of detection" (lines 182 f.).**

Line 148 The surface lowering is relatively more active in the 3-week period, but you also show the total amount of lowering is much lower that the amounts of accumulation – you detect mass accumulation and thickening overall, so caution should be exercised in the interpretation of these results, it might help to put these rates in absolute terms.

We added mean absolute rates and adapted the phrasing to make this element more concise:

"Interestingly, the mean positive surface changes are 0.04 m in the three-week period (one-year period: 0.27 m) and the mean negative surface changes are -0.03 m (one-year period: -0.14 m). The contribution of the three-week period to the annual positive surface change in the normal direction amounts to 14.8 %, while this ratio is 21.4 % for negative surface change over the same point set. The higher proportion of negative change indicates that apart from the dominant process of mass accumulation and thickening affecting both periods, surface lowering is more active over the three-week period than surface raising." (lines 186 ff.)

Line 149 what is the difference between 'rates made uniform over the year' and 'average across the year'? Furthermore, this assumption is clearly not appropriate for an alpine rock glacier which we would not expect to record uniform rates through a yearly cycle.

The aim here is to show in simple terms that the rate of surface lowering during the three-week period is clearly above-average. We also want to highlight that a lot of detail is missed when change rates are averaged over a year due to a large survey interval. To point this out more clearly, we improved the phrasing to:

"The rate of surface lowering is four times what would be expected if changes observed over the annual period were uniform through the year or, critically, if variable changes were averaged across a year by the user due to an annual survey interval (5.7 %)." (lines 192 ff.)

Line 160 – 161 This is rather confusing. As I understand it, your 'orographic left' would be the reader's right? In this area I can't see 'small and discrete areas of positive and negative change' that you infer as 'boulder movement' but rather large (nearly half of zone 1) coherent blocks of negative (upslope) and positive (downslope) change. The change is very patchy throughout the annual change, particularly on the margins, is this registration error between scans? It seems strange to have reversed patterns of change on either side of the slope with accretion high upslope and lowering downslope in patches on the left of image 4b and the opposite in the right of zone 1, more explanation is needed. Again perhaps combining with drivers of change or underlying bedrock might help.

The readers' left corresponds to the orographic left (flow direction is appr. North). We realize the ambiguity in this paragraph and have included an arrow showing the direction of creep in Fig. 4 to clarify.

We interpret the mentioned coherent blocks of negative and positive changes to stem from a large chunk of material that has melted out of the rock glacier body and slid downslope. They are not addressed in the paper, but we agree that such events and processes would be very interesting to investigate with more in-situ data in future campaigns.

Line 189 I'm not sure the movement data in 5a are 'more homogenous', the directions seem to differ significantly, perhaps consider rephrasing around localised but coherent directions. It seems evident that independent boulder movement must have occurred but just haven't been detected, as you point out how difficult this is in the annual data in Fig. 3b – it should be framed in this context rather than 'cannot assume they didn't'.

**We changed the section to:**

"This is indicated by magnitudes of boulder movement that are all well below 5 m (Fig. 5b). Boulders in the one-year period moving independently from the rock glacier advance (presumably over distances > 5 m) could not be identified visually, because they likely could not be re-identified between

successive point clouds due to the many occurrences and large distances of boulder movement." (lines 231 ff.)

Line 193 What does a 0.10 m contribution to -0.5 m actually mean? Do you mean a -0.1 m contribution or did it offset the negative change by 0.1 m?

**We corrected this typo to "-0.1 m" (line 238).**

Line 234 What do you mean by an 'increase in surface velocity'? Make sure it reconciles against the final line of the results 'movement was in-line with the annual average'.

Based on your comment, we acknowledge that the general surface velocity increase during warm summers is not relevant for our argument regarding thaw settlement. We therefore changed the sentence to:

"Long-term measurements of cross-profiles on the *Äußeres Hochebenkar* rock glacier (Hartl et al., 2016) have shown that warm summers with high precipitation can lead to a decrease in surface velocity at the lower end of the tongue, indicating an ice loss due to high temperatures and a subsequent velocity decrease." (lines 284 ff.)

Line 248 - 255 This is the first we hear of really valuable site data. It seems a shame that this was not introduced earlier and made much more use of. Surely it is relevant to the site context and selection of zones and then interpretation of results (note the comment above on Fig. 3 for example) – and much more could be made of the spatial patterns identified if better contextualised with this previous data.

For context, we inserted the existence of the GPR data into the study site section (lines 87 f): "Ground penetrating radar (GPR) has been applied to determine the depth of the bedrock, with estimates of mean thickness between 30-40 m (Hartl et al., 2016a)."

We pick up on these external site data in the discussion to put the merit of our results into context with existing geomorphic research of rock glacier dynamics. As discussed above our main focus of this study is the development of the method to derive input (including datasets, how to set up the system and perform the 3D data analysis) for those follow-up studies analysing the drivers and increasing the understanding of the site-specific mechanisms.

**Technical corrections**

Strictly speaking raster based change detection is not 2D as inferred in line 42-3, they are typically able to identify surface change in X, Y, and Z directions, and should probably be referred to as 2.5D to distinguish from full 3D analysis because the z component is assumed normal to the raster X,Y grid (as noted in Williams et al., 2018).

**We changed the statement to "2.5D" (line 46).**

Line 67: Reword 'This will enable to disclose' to better grammar.

**We rephrased the sentence to:**

"The contribution of short-term surface changes to annual surface changes will be derived for the first time, here as a ratio between the surface change occurring during a three-week period and that over one year." (lines 71 ff.)

**We corrected this accordingly (line 182).**

Line 171 'move creep-induced' needs rephrasing

We changed the phrase to "that their movement is induced by creep" (line 214).

Line 268 Should read 'at a sub-monthly interval' as you on used one three-week period.

We adapted the phrasing as suggested (line 320).

Figure 1

It's unclear what the 'three epochs of TLS point clouds' are? Provide dates, as it is unclear how these relate to the changes claimed until later in the paper or switch the order and present it after line 84 and it possibly should be in the Methods section anyway. It is also unclear how you know the monitored 3-week period is one of 'heightened activity'? Is this written in retrospect or do you have more data and have selected this 3-week period as that of heightened relative activity?

The figure has been inserted into the methods section and is now Fig. 2. We have provided the dates and removed 'heightened activity' from the figure as it can indeed not be known before the analysis of the given data.

Figure 2

An inset map for context would help locate the study.

We inserted an inset map showing the location of the study site in Austria (now Fig. 1).

Fig. 3 Minor point, but it makes more sense to have the wider context and first survey done first – so I suggest swapping a and b around. Where is this profile on the slope? The movement of the boulder is intriguing, appearing to rise up and over a slope crest, is there an underlying ridge present? Some indication of the rock base would be very useful if available (I appreciate it may not but perhaps inferences could be made from the adjacent ridgelines). State 'Level of detection (LOD)' as you use it in the figure. (4) appears to be missing? Also I'm not sure Z – displacement is the correct title for the Y-axis as you have expressly stated M3C2 distance is not just in the Z direction, I would advise this should just be 'Distance'.

We swapped "a" and "b" in the figure to have the wider context first and to have the numbering of the different processes in the figure in a logical order. Fig. 3 is not intended as example of processes occurring at a certain location, but a schematic representation showing the types of surface change that can be observed on the rock glacier. It is therefore not possible to interpret any specific underlying processes from this figure.

We changed the axis labels to "Vertical Distance" and "Horizontal Distance" and added the number 4, corresponding to "Surface Movement

[revised manuscript text omitted]
, Wwe examine the benefits of interpreting\_-3D surface changemovement at sub-monthly intervals in relation from short interval a sub-monthly interval monitoring in terms of its contribution toas a function ofto annual movement, here in the context of superimposed, and hence cumulative, surface changes and the drivers behind it. By We aim to quantifying surface change based on movements that occur in different directions: <del>processes of surface change (e.g. surface lowering or surface heave)</del>. To achieve this, movements normal to the surface of an
- 70 active rock glacier, derived from the M3C2 algorithm, and movements in the direction of rock glacier flow, derived from individual boulder tracking, will be used (Fig. 1). The contribution of short-term surface changes to annual surface changes contribution of a three week period to the annual surface change will be assessed derived for the first time, here by considering this contribution as a ratio between the surface change occurring during the a three-week period and the surface changethat within over one year. This will enable . Following this approachAs a result, it will be possible to disclose the contribution of short-term surface changes to annual surface changes in the context of superimposed, and hence cumulated, surface changes

2 Study Site and Data

for the first time.

[revised manuscript text omitted]

---

## Author Response (AR2)

Dear Dr. Schwanghart,

thank you for your decision and additional feedback. We are very happy to address your final remarks. Please find our detailed response below with reference to revisions in the manuscript (line numbers refer to tracked changes document).

Please do not hesitate to contact us in case of further issues,

Bernhard Höfle and Veit Ulrich on behalf of all authors

TITLE: Your analysis relies on two main techniques: Quantification of surface change relative to the surface normal using the M3C2 algorithm, and the tracking of boulders. Both techniques measure surface changes but not necessarily different surface change types. The term "disaggregating" in the title suggests to me that one technique is used to correct for a surface change type that potentially influences measurements made by the other technique. For example, the flow field measured by boulders could potentially be used to dissagregate surface changes to those invoked by creep and those due to frost heave or thaw settlement. I am not seeing where this is actually done in this study. However, I am not demanding that it should be done. Rather, I'd suggest to reformulate the title to better reflect the focus of this study, which is, in my view, the comparison of three-week vs one-year observation period.

We changed the title to "Measurement of rock glacier surface change over different timescales using terrestrial laser scanning point clouds"

FIG 3: Here you use the abbreviation LOD which you should mention in the caption, too. You might consider using this abbreviation throughout the paper, too.

We have added the abbreviation in the caption (line 127) and are now using it throughout the paper after introduction in line 138 f.: "… referred to as the level of detection (LoDetection)"

137: least squares fit?

We have specified the method in the sentence accordingly (line 139 f.): "This draws on (1) the least squares fit of points within each neighborhood to a plane, with a higher standard deviation of residual distances …"

138: residual distances?

We have adapted the phrasing accordingly, see edited text in previous comment (line 140 in the manuscript).

147: consider adding some information about the size of the boulders.

We now provide information about the diameters of the boulders:
"In these active zones, the movement of 48 manually identified boulders with diameters ranging from 0.3 m to 1.1 m was measured in both observed periods." (lines 150 ff.)

159: consider providing the exact number of boulders here.

We have added the exact numbers:
"Here, the correspondence between both epochs could be verified visually for a limited number of boulders only (eight in the three-week period and seven in the one-year period)." (lines 161 ff.)

FIG 5: It is a bit confusing that panel a and b show different observation periods and places. Would it be possible to place both in one panel and indicating the different periods using different colors for the arrows?

Placing both parts of the figure in one panel would require reducing the scale considerably due to their distance within the extent of the rock glacier (cf. Fig. 5c). This would make it impossible to display particularly the shorter arrows, which are to scale with the movement. We therefore prefer to keep the figure in its present form. However, we have added the labels indicating the time periods into the subfigures.

213: I am missing a figure that shows the boulder tracking results for the active zone 2.

We have inserted a figure showing the boulder movements in active zone 2 (Fig. 6). It shows that the boulder movements on the rock glacier surface induced by creep are more homogeneous regarding distance and direction than the boulder movements in active zone 1. We have renamed the previous Fig. 6 to Fig. 7 and adapted all figure references in the text. To incorporate the figure into the text, we have added the following sentences:

[revised manuscript text omitted]